# Interdependent Autonomous Human–Machine Systems: The Complementarity of Fitness, Vulnerability and Evolution

**DOI:** 10.3390/e24091308

**Published:** 2022-09-15

**Authors:** William F. Lawless

**Affiliations:** Departments of Mathematics and Psychology, Paine College, Augusta, GA 30901, USA; w.lawless@icloud.com

**Keywords:** entropy, interdependence, complementarity, autonomy, human–machine systems

## Abstract

For the science of autonomous human–machine systems, traditional causal-time interpretations of reality in known contexts are sufficient for rational decisions and actions to be taken, but not for uncertain or dynamic contexts, nor for building the best teams. First, unlike game theory where the contexts are constructed for players, or machine learning where contexts must be stable, when facing uncertainty or conflict, a rational process is insufficient for decisions or actions to be taken; second, as supported by the literature, rational explanations cannot disaggregate human–machine teams. In the first case, interdependent humans facing uncertainty spontaneously engage in debate over complementary tradeoffs in a search for the best path forward, characterized by maximum entropy production (MEP); however, in the second case, signified by a reduction in structural entropy production (SEP), interdependent team structures make it rationally impossible to discern what creates better teams. In our review of evidence for SEP–MEP complementarity for teams, we found that structural redundancy for top global oil producers, replicated for top global militaries, impedes interdependence and promotes corruption. Next, using UN data for Middle Eastern North African nations plus Israel, we found that a nation’s structure of education is significantly associated with MEP by the number of patents it produces; this conflicts with our earlier finding that a U.S. Air Force education in air combat maneuvering was not associated with the best performance in air combat, but air combat flight training was. These last two results exemplify that SEP–MEP interactions by the team’s best members are made by orthogonal contributions. We extend our theory to find that competition between teams hinges on vulnerability, a complementary excess of SEP and reduced MEP, which generalizes to autonomous human–machine systems.

## 1. Introduction

Our objective for this work-in-progress is to develop mathematical physics models of autonomous human–machine teams and systems (A-HMT-S) to guide human–machine systems. Our success thus far began by attributing the social sciences replication crisis [1] to a science based on independent individuals [2]; this resulted in “causing” many static theories in social science to not only become invalid (e.g., self-esteem [3]; the implicit attitudes test [4]; ego depletion theory [5]), but also non-generalizable to theories of human–human interaction, let alone to autonomous human–machine teams and systems (A-HMT-S).

To better illustrate this problem in social science, the implicit attitudes test (IAT) has become widely used at great expense to industry, even though the concept is invalid [3] and its interventions ineffective [6]. Later, the lead scientist on the team that found the IAT to be invalid, Tetlock, wrote a book about training practitioners to become “superforecasters” [7]; Tetlock started a public website with challenges for his hand-picked and “trained superforecasters” to make predictions; his first two superforecasts were that Brexit would not happen and that Trump would not become President of the U.S. (http://goodjudgment.com/superforecasting/index.php/2016/11/03/is-donald-trump-mr-brexit/; replaced with gjopen.com (accessed on 15 December 2016)); both predictions failed. Additionally, of note, the leader of the group that found self-esteem to be invalid, Baumeister, produced ego-depletion theory, which became one of the leading theories in social psychology [8]; it is a concept that has since been found to be questionable [5].

Machine learning has limits, too. Machine-learned intelligence is non-generalizable, e.g., [9] reports that “current machine learning systems typically cannot identify differences in contexts, let alone adapt to them,” likely due to machine learning’s “unreliable empirical risk minimizer” on known sets of training data [10]. Machine learning programs trained with big data are satisfactory for known, stable contexts [11], but not for new, uncertain, or dynamic contexts (e.g., social contexts), where the failure of big data has been attributed to its being independent and identically distributed (i.i.d.) [12], precluding Shannon information from being able to reconstruct whatever social event is being observed.

To counter these problems and to build the science of autonomy, investigators have begun to call for a new direction [12]. Echoing this call, the National Security Commission on Artificial Intelligence (NSCAI; in [13]) reported that the “profound” changes being wrought by AI (and machine learning) posed technological concerns that had to be addressed to prevent war and to safeguard freedom. To prepare for autonomy, in this article, we address these calls for a new direction and some of the concerns raised about AI. We propose that the new direction must address environmental uncertainty, and we explain in this study why models based on logic cannot. The National Academy of Sciences provided support for this new direction by reviewing the research gaps in this new field, by reviewing interdependence and by addressing the need for open science research conducted in the field as opposed to laboratory research [14]; we cover in this study: autonomy; uncertainty; and open science field research.

## 2. History: Complementarity and Interdependence

Bohr [15] noted for “the ideal of causality, that the behavior of a physical object relative to a given system of coordinates is uniquely determined, quite independently of whether it is observed or not” (p. 290). Causality ended at the atomic level, however, when Heisenberg’s uncertainty relations for an object’s pairs of conjugate factors found that one predicted outcome could be known precisely but not with the other simultaneously [15], e.g., position–momentum; energy–time. Bohr wanted to generalize Heisenberg’s principle of uncertainty. He learned that William James, the psychologist, distinguished between a concept and its expression: “consciousness may be split into parts which coexist but mutually ignore each other, and share the objects of knowledge between them. More remarkable still, they are complementary” ([16], p. 204).

With this idea, Bohr began to replace causality with complementarity to reflect dependence between a human actor and observer. According to Bohr, the dependency in complementarity occurs when an object interacts with a measurement instrument, that is, when they become interdependent. Bohr [17] generalized complementarity to the behavior of individuals; to actors and observers; and to Culture *A* and Culture *B*. In these and other relationships, Bohr used the term complementarity to describe interdependent elements that combined concepts and behaviors. He concluded that these dependent roles, say of an actor and an observer of the actor, both collect different information from reality.

For this new direction, there is research in social science that can help. One theory in social science that is generalizable separates team structure [18] and performance [19,20]. Another generalizable social concept, borrowed from Schrödinger [21], from Systems Engineers [22] and from Social Psychologists ([23], p. 146), is the claim that the whole is greater than the sum of its parts. Schrödinger [21,24], the quantum theorist who discovered entanglement, concluded that dependency in the parts interacting inside of a whole meant that knowing a whole with dependent parts was possible, but not its parts simultaeously. From our perspective, if the parts reflect autonomous, interdependent humans acting in orthogonal roles, the information that they generate may not correlate, contributing to the replication crisis, and yet, advancing the science of complementary interdependence [20].

We present evidence in this article that rational models derived from individuals fail when exposed to conflict and uncertainty [25], whereas complementarity is effective and generalizable.

## 3. Background: Why Team Fitness and Entropy?

The motivation for human–machine teams and systems is being driven by the need to make decisions faster, more accurately and more safely than can humans alone. Examples exist across the whole of society, with the arrival of autonomous transportation [26], the introduction of autonomy into mundane banking and advanced finance processes [27], the sophisticated applications of “medical and industrial robots, agricultural and manufacturing facilities, traffic management systems” [28], and their introduction into complex weapons systems such as hypersonic missiles (e.g., [29]). However, for weapons, the editors of the *New York Times* [30] warned of the threat to humanity from errant autonomous systems, citing the UN Secretary General, Antonio Guterres, who stated that “machines with the power and discretion to take lives without human involvement … should be prohibited by international law.” The Editorial recommended that “humans [should] never completely surrender life and death choices in combat to machines.”

Our response to the *New York Times* exemplifies the interdependence found in every aspect of society. Interdependence has long been a part of the human condition, a thread running through diplomacy (e.g., Henry Kissinger), economics (Adam Smith), the law (Richard Freer’s law student casebook), war (von Clausewitz and Emile Simpson) and governance (Kessinger, Montesquieu and Madison). For example, according to Kissinger [31], “diplomacy is the art of restraining power.” In Smith’s [32] view of business, the economic wealth of a nation is promoted when its citizens are free to pursue their own interest in their own way: “It is not from the benevolence of the butcher, the brewer, or the baker, that we expect our dinner, but from their regard to their own interest.” Freer and Purdue [33] write that justice in the courtroom is found from the arguments of equally competent lawyers. According to von Clausewitz [34], “war is not merely a political act but a real political instrument, a continuation of political intercourse, a carrying out of the same by other means” (p. 280); Simpson [35] adds that war is a competition to impose meaning (p. 35). For the least corrupt form of governance, according to Kissinger ([31], pp. 18, 21), the “realities of interdependence” self-organize the balance of power shared among nations. Within a nation, Montesquieu‘s concept of checks and balances to reduce corruption was adopted by Madison [36] so that “ambition must be made to counteract ambition,” a concept now enshrined in the U.S. Constitution. These examples of interdependence provide the backdrop for our response to the Editors of the *New York Times*: What about a machine that can interdependently counter a copilot who might be intent on committing suicide, or interdependently guard against a distracted train engineer or a negligent crew aboard a ship? Should we not permit the most capable agents, human or machine, to save humans under threat [37], even when that mechanism is a machine? An example of this is the case of unconscious fighter pilots saved by their AI: https://futurism.com/the-byte/fighter-pilots-blacked-out-saved-ai (accessed on 15 March 2021).

What we have established at this point is that social science is not generalizable to autonomous human–machine teams and systems and that a similar problem exists with machine learning, but that human life entails interdependence in every context. Yet, these new teams and systems are soon to arrive. How to construct them and measure their performance are critical questions. Part of the problem with social science is that it has relied on questionnaires, interviews, observations and other measurements that interact with the individual humans being observed [20]. Like the measurement of quantum objects, we have known for decades that interdependent social objects are affected when they are being observed (i.e., the Hawthorne effect [38]). In contrast to the traditional social science of individuals, we propose two new measures for teams: structural entropy production (SEP) and maximum entropy production (MEP). We shall argue that these two entropy measures offer a more objective means to determine the performance of autonomous systems.

The plan for this manuscript: next, we will review the theory of complementarity, interdependence, the mathematics and its relationship to entropy and our Research Design, Results, Discussions and Conclusions.

## 4. Theory

Entropy is a thermodynamic quantity in a system that reduces the system’s available energy to perform useful work [39]. We will argue that the structure (the configuration, the skills and roles of team members, etc.) of a team are complementary to its performance, that the more available energy expended to make a team’s structure effective, the less available energy the team or system has to perform useful work.

Trusted artificial intelligence (AI) in human–machine teams places more emphasis on a solution than the science of individuals can support [14]. We first addressed this issue in a book chapter [40] followed by an Editorial in a Special Issue of *AI Magazine* [41], leading us to conclude that a context with uncertainty can only be determined interdependently by intelligent teams in competition or conflict. In causal terms understood by machines and humans, Pearl [42,43] recommended that human–machine teams must be able to communicate interdependently. In 2018, the concept of team interdependence was violated by Uber’s self-driving car when it fatally injured a pedestrian even though the car had discovered an object in the road 5 s before its human operator, but the car, acting independently, did not communicate that information to its human teammate [24,44]. In contrast, rejecting interdependence, Kenny and colleagues ([45], p. 235) assumed that only independent, identically distributed data are of value for experimental social data, which is somewhat akin to treating quantum dependency at the atomic level as an irritation, not a resource; we have attributed Kenny’s assumption to the root cause of the failure of replication in the social sciences, and its inability to generalize to teams and machines [46].

We know that beliefs correlate with other beliefs (e.g., the beliefs of individuals on their self-esteem correlate significantly with beliefs in their academic and work abilities); however, self-esteem does not correlate with actual academics or work performance (reviewed in [20]). In sum, we need a theory that accounts for the failure of observational social science (e.g., self-esteem) to predict behavior; for the failure of behavior (e.g., reinforcement learning) to predict cognitive states; and for the failure of social scientists to produce a theory that accounts for both. Without this knowledge, how else are we to theorize, design and operate human–machine teams?

## 5. Interdependence: Structure, Performance and Uncertainty

We have associated interdependence with three effects: bistable perspectives of reality, such as those that spontaneously arise between two tribes, teammates, or political parties [11]; a measurement problem arising with the collapse of bistability; and non-factorability, the difficulty of unraveling the causes of social conflict or competition [20]. We argue that the ignorance of these effects contributes to the replication crisis in psychology.

First, interdependence is entailed in all social interactions [47], the original premise of game theory [2]. Conant [48] generalized Shannon to organizations, concluding that interdependence should be minimized. Contradicting Conant, Cummings (in [46]; also, see [49]) found that an interdependent team is significantly more productive than the same individuals in the team when they are working independently. Moreover, contradicting the National Academy of Sciences’ claim that “more hands make light work” ([50], p. 33), and in contrast to swarms, we propose and found—based on studies on international oil firms [2], replicated with militaries around the globe [51]—that redundancy in an interdependent team impairs its performance, e.g., in 2018, although both companies produced about the same amount of oil, Exxon had one eighth the number of employees than had Sinopec [52].

Second, interdependent states include bistable interpretations of reality [53], which are spontaneously evident when uncertainty in a team prevents action. The brain only “interprets” one aspect of bistability at a time [54], where bistable refers to multiple interpretations, e.g., a bistable illusion is the faces–vase illusion (see p. E3351 in [55]). The split-brain evidence from Gazzaniga [56] showed that the two halves of a human brain perceive independent versions of reality. However, the brain apparently stitches these independent visual memories from the hemifields of the left and right brain together [57], possibly by managing constructive and destructive interference from the information produced by the brain’s two halves as the brain constructs a coherent view of reality. Bistability supports Bohr’s claim of the complementarity between action and observation, tribal disagreements, etc. The power from bistable interpretations also supports the claim that interdependence between society and technology drives the evolution of humans [58].

How this stitching of “reality” occurs in the brain’s “mind’s eye” is unknown, but may depend on the development of a new theory of information value; similarly, as we will discuss, we do not yet know how political information is aggregated beyond the realm of logic or our present knowledge of the process. We speculate that both are related to the theory of the non-factorable value of information, which we leave to a future project.

Third, indeed, many aspects of social reality are non-factorable, requiring an adversarial process to de-construct interpretations of reality, e.g., a high-profile child custody case; the NTSB’s determination of responsibility for the Uber pedestrian fatality in 2018 [24,44]; and Weinberg’s failure to reach consensus about the meaning of quantum mechanics (compare [59] with [60]). Humans pursue adversarial struggles to determine the non-factorable reality hidden in uncertain contexts. As non-factorable information is processed in an adversarial process, it tests the bistable interpretations of an uncertain reality. For example, Justice Ginsburg ([61], p. 3) concluded that the lengthy appeals process for a case reaching the U.S. Supreme Court should not be short-circuited so that the Justices can best gain an “informed assessment of competing interests.” Ginsburg’s decision exemplifies the processing of interdependent (bistable) information, and the brain’s “mind’s eye” appears to function interdependently to produce a similar effect.

Interdependence is fragile and has to be contained inside of stable boundaries to preclude interference; boundaries suggest an engine, a dual function of the constraints to stabilize the interaction, to block it from interference and to allow the interactions of a team to coherently increase the team’s performance to reach its maximum entropy production (MEP) [19,20].

Subsequently, in an earlier study using UN Human Development Index (HDI) data for Middle Eastern North African (MENA) nations plus Israel, mostly the data on schooling and patents produced, we theorized and found that a nation’s structure of education is significantly associated with a nation’s MEP, characterized by the number of patents it produces; in this manuscript, we revisit these results. However, our result conflict with our earlier finding that a United States Air Force education in air combat maneuvering was not associated with the best performance in air combat, but air combat physical training was. These two results, to achieve MEP, combine to exemplify that two different structures built to exploit the interactions in a team are not based on independent individuals, but on interdependent teammates who function in orthogonal roles [51].

There are two basic approaches to social science, both occurring most often in the laboratory: behavior and cognition. First, with the focus on fixed choices, games can impede a “proof of concept” in the field for operating human–machine teams [2]. Briefly, games work by assuming that the preferences of players are implicit based on observed behaviors (viz., game choices; in [62]). Von Mises makes the same assumption in his theory of entrepreneurship [63]. Second, and in contrast, cognitive psychology is based on implicit behaviors [64]. The implicit assumptions in both traditional disciplines help to make these concepts rational. Both approaches function well within the limits of their assumptions, but when it comes to generalizations for operations in the field, a new direction is needed in the situations where uncertainty and conflict operate; this is exactly where rational theories fail, despite heroic effort [25]. Namely, the lack of field research, also known as “open science,” is the largest gap in research reported by the National Academy of Sciences. The Academy blamed this gap on the emphasis by social scientists on conducting research in the laboratory that often cannot be generalized and applied in the field [14].

For example, a modern fighter jet is “trained” to take control from its human operator to save the pilot and plane when the machine (plane) recognizes that its fighter pilot’s high “g” maneuver has caused its pilot to suffer a loss of consciousness, known as G-LOC. Similar technology can be applied to counter dysfunctional human operators, e.g., an attempt to commit suicide by a commercial pilot can be countered by allowing a plane to safe itself and its passengers (e.g., the Germanwings copilot who committed suicide and killed all aboard in 2015 [20]).

More applicable in the field is a model that performs similarly to how autonomous humans combine behavior and cognition to process the information interdependently generated during competition, by testing whatever implicit assumptions arise. A foundational problem for traditional social science, we have found that the theory of independent individuals, not surprisingly, does not scale from individuals. In contrast, our theory not only scales from teams to firms and systems, but also scales downward from teams to individuals [20,46].

## 6. Mathematical Physics of Complementarity

When two agents or two groups, *A* and *B*, are independent, we can represent the information derived from them using Shannon entropy in Equation (1) [65]:(1)$$S_{A,B}\geq S_{A},S_{B}$$

Alternatively, when the two agents or two groups are fully independent, we revise Equation (1) to become:(2)$$S_{A,B}=S_{A}+S_{B}$$

Equations (1) and (2) represent a state of independence; based on this, the recommendation from computational Shannon models is to minimize interdependence [48]. Equations (1) and (2) identify a lack of interdependence and reflect either social independence, or the breakdown of a preexisting state of social interdependence. In the former case, such a situation could reflect completely unrelated or independent activities; in the latter situation, Equation (2) could reflect social suppression [24], e.g., the Stasi secret police in East Germany made individuals fearful of divulging personal information, creating forcible isolation among East German individuals [66].

From Cummings [67], however, we know that the most coherent and productive teams of scientists are highly interdependent [67]. In contrast to Equations (1) and (2), we assume that a state of dependence with one agent’s dependence on the other is sufficient to reduce its contributions to their joint entropy, but that this reduced state forms a state of interdependence or mutual dependence, e.g., representing a team (a business unit, a marriage, or a boss–employee relationship). In these cases, the degrees of freedom are correspondingly reduced for teams [24]. Further, in the limit, as a team forms structurally into a unit, this reduction produces a joint entropy, $S_{A,B}$, that, in theory, can become zero for a perfect team:(3)$$S_{A,B}=lim_{\Sigma n\left(individuals\right)\to team\to \mathrm{dof}\to unit}log\left(dof\right)=0$$

When this lowest state of structural entropy production occurs, because the entropy produced by one structural part of a team is offset by another structural part of the team, at this point, the entropy becomes sub-additive, and is reflected as:(4)$$S_{A,B}\leq S_{A}+S_{B}$$

Another interpretation of Equation (4) shows, counterintuitively, that the structure of the whole is producing less entropy than its parts:(5)$$S_{Whole}\leq \Sigma_{i=1}^{n}S_{i}$$

In Equation (5), the summation of the structural entropy occurs over all of the elements, *i*, or parts of the whole, from 1 to *n,* where *n* represents the whole team. For the first time, Equation (5) not only reflects Lewin’s [23] claim, but also the claim by Systems Engineers [22] and the assertion by Schrödinger that the whole has become greater than the sum of its parts [21]. This result occurs when the parts of a whole contribute to the performance of the whole, and each part is dependent on the other, precluding an objective determination of each part’s contribution. We found support in the assertion by the National Academy of Sciences, in their recent report, that the “performance of a team is not decomposable to, or an aggregation of, individual performances” ([14], p. 11). Equations (4) and (5) account for this assertion.

However, what about generalizing this result to two teams? We address this question with a simulation. If we set *b* as the belief in a concept to solve a problem that increases the negative entropy (negentropy) as the belief circulates in a series circuit against the social resistance, *R* (which impedes the acceptance of a concept) the social capacitance, *C* (which acts to store the energy behind opposing beliefs) and the inductance, *L* (which creates a field to resist the directive flow of energy for a concept, inducing a concept’s flow in the opposite direction when the field reverses), we obtain:(6)$$Lb+R+\frac{1}{Cb}=0$$
when *L* = 1 unit, *R* = 0 units, and *C* = 1 unit, we obtain *b* = ±i for points 1 and 2 on the *y* axis in Figure 1. When *R* = 2 units, solving for *b* gives −1 and −1 for points 3 and 4 on the *x* axis in Figure 1. When *R* = 3 units, *b* = −2.6 and −0.4, reflected by points 6 and 5 in Figure 1.

Figure 1 simulates a confrontation between two individuals or two teams in an argument that cycles back and forth (e.g., a competition, a debate, or collaborative decision-making) that precedes and may be followed by a decision to act (action is more likely to occur under majority rules than consensus-seeking rules; for an example in European matters, see [69], p. 29).

The Department of Energy’s (DOE’s) cleanup of its nuclear waste mismanagement best represents Figure 1 [70]. As an example of a pure oscillation, in 2005, the DOE’s high-level radioactive waste (HLW) cleanup of its tanks had been stopped by a lawsuit, but was restarted by U.S. Congress. As part of this new law, the U.S. Nuclear Regulatory Commission (NRC) was given sufficient oversight to overrule the DOE’s technical decisions for its HLW tank closure program. However, from 2005 to 2011, the DOE proposed a restart of its HLW tank closure program, but NRC equivocated, then required the DOE to make another proposal. This pure oscillation is represented by the back-and-forth between points 1 and 2 in Figure 1. The back-and-forth, approaching 7 years, lasted until the DOE complained to its Citizens Advisory Board that it was in danger of missing its legally mandated milestone to restart its tank closures. The citizens demanded, in public, that the DOE and NRC settle their differences and immediately restart tank closures, which happened, reflected by point 6 in Figure 1. If we let $\left(\begin{array}{c} 1\\ 0 \end{array}\right)$ represent 1 on the *x* axis and 0 on the *y* axis, we simulate the rotation in the minds of the NRC, shown by Equation (7), as an orthogonal rotation of 90 degrees:(7)$$\left(\begin{array}{cc} cos\;\theta & -sin\;\theta \\ sin\;\theta & cos\;\theta \; \end{array}\right)\left(\begin{array}{c} 1\\ 0 \end{array}\right)=\left(\begin{array}{c} \cos\pi /2\\ \sin\pi /2 \end{array}\right)=\left(\begin{array}{c} 0\\ 1 \end{array}\right)$$

In Figure 2 below, we plot a simplified simulation of the NRC’s decisions over time.

We simulated the DOE–NRC stalemated oscillations using an online five-step Markov chain and transitions initially set at 0.6 from State 0 to 1, etc., which we changed to create an oscillation from State 0 to State 2. With game theory played online, we re-ran the DOE–NRC standoff for almost 7 years as a Nash equilibrium. A Nash equilibrium occurs when neither side is a winner, or when both sides win. We conceptualized the DOE–NRC standoff as a quasi-Nash equilibrium because neither side won; however, the American public was losing until the DOE’s Citizens Advisory Board was told about the standoff, and then, confronted both Federal agencies in public and demanded that they make a decision to allow the tank closures to resume [70]. The tank closures resumed almost immediately in what one DOE official described as the fastest action he had ever witnessed in his career.

To summarize, the DOE–NRC situation is represented in Figure 1 and plotted over time in Figure 2. Figure 1 reflects harmonic motion where beliefs are exchanged but no decision to act is made (on Figure 1’s imaginary axis, an oscillation is between points 1 and 2, then 1, etc.), resulting in a compromise won by neither agency (points 3 and 4 on the real axis), or in resistance by the audience and a decision it has forced on the agencies, in this case (points 5 and 6).

Competition. In reviewing the literature (e.g., von Mises, in [63]), competition is often described as a struggle between two or more organisms, humans or businesses to be more efficient [32]; to market a better product or an existing product more quickly; to win a position or grant (as in education); or from ecology, “in which birth, growth and death depend on gaining a share of a limited environmental resource.” (https://www.lexico.com/en/definition/competition (accessed on 26 June 2022)) However, in the many and varied definitions of competition, the “how of competition” is often omitted, ignored or assumed to have happened. Shifting to autonomous human–machine systems makes intuition unsatisfactory. To rectify this situation, we build on the rotations represented by Equation (7) to introduce a decision advantage, *DA*, with torque, τ; force, *F*; the radius of a turn, *r*; and rotational velocity, ω; to give τ=rω=Fr; then, with power, *P*, we obtain: $P=Fv=\tau \omega =\tau \left(v/r\right)$, becoming [40]:(8)$$DA=\tau_{A}/\tau_{B}$$

In Equation (8), we mean by *DA* that one team was quicker than another in its oscillations in the back-and-forth between competitors during a debate, or that one team’s grasp of the issues was more forceful, or that one team’s knowledge of its solution was held with more conviction than the other.

Equation (8) has support in the literature and also in the field. According to McConnell [71], the Director of National Intelligence, “… strategic advantage is the ability to rapidly and accurately anticipate and adapt to complex challenges ([71], p. 6). … the key to intelligence-driven victories may not be the collection of objective ‘truth’ so much as the gaining of an information edge or competitive advantage over an adversary” ([71], p. 8) (see also J. Sims, Director of Intelligence Studies, Georgetown University, quoted in McConnell [71]). In this environment, one prerequisite for decision advantage is global awareness: the ability to develop, digest and manipulate vast and disparate data streams about the world as it is today. Another requirement is strategic foresight: the ability to probe existing conditions and use the responses to consider alternative hypotheses and scenarios, and to determine linkages and possibilities ([71], p. 9). Secrecy, however, is only one technique that may lead to decision advantage; so may speed, relevance or collaboration ([71], p. 9). 

The purpose of decision advantage in combat, or in preparation for it, is to “exploit vulnerabilities” ([72], p. 7). Speed and quality decisions are important in business, too [73]. Additionally, the same is true in advertisements that promote athletic performance (e.g., [74]).

In Equation (5), we reported on how the whole is greater than the sum of its parts for a well-functioning team. However, internal conflict is all-too-common, reversing the relationship between the whole and its parts. In the literature, business “Executives underestimate the inevitability of conflict, but also … its importance to the organization” [75]. In the field, in the *Wall Street Journal* [76], after J&J discovered that Emergent had ruined a batch of COVID-19 vaccines, J&J replaced its Emergent BioSolution contract manufacturer “in response to the recent discovery of the contaminated batch.” According to the *New York Times* explaining that a candidate faltered because of internal conflict [77]: “Dianne Morales, a former nonprofit executive, once appeared poised to be a left-wing standard-bearer, but her standing suffered amid internal campaign turmoil …” Another example from the *Bangkok Post* about the destruction of the newspaper business in an attempt to save it [78] states that “As a major Tribune shareholder, Alden backed cuts last year at the company—amid a steep, pandemic-driven advertising downturn—that slashed a third of the total payroll and closed about half its news offices, with no stated plans to open new ones.”

In these examples and numerous others (divorce; forced business spin-offs; hostile mergers; war, etc.), the whole may be significantly less than the sum of its parts, which we represent in Equation (9) to show counterintuitively that the whole is now superadditive, producing more entropy than the sum of its parts:(9)$$S_{whole}\geq \Sigma_{i=1}^{n}S_{i}$$

For the purpose of illustration, in Figure 3 below, we show how we integrated these two concepts for the two Equations (5) and (9).

From the discussion, we next assume that the ability of a team to be productive is related to its ability to minimize the team’s configuration entropy produced by its structure [46]. We show that achieving minimal or the least entropy production through a team’s structure (SEP), as suggested by Equations (4) and (5), allows a team to direct more of its available free energy to increasing its productivity [39], possibly even to achieve maximum entropy production (MEP; [19]). An example of MEP would be a highly productive team, e.g., a marriage devoted to raising its children; a business such as Apple Technology Company becoming ever more productive in its struggle to reach MEP based on its structure; an alliance devoting all of its available free energy to solving the problems placed before it; or a nation’s people educated to search through potential solution spaces to solve the problems faced by a nation with solutions that can be patented [52].

Finally, we assume that uncertainties for both the structural entropy production in Equation (4) and the maximum entropy production in a highly productive team are interdependent. We conceptualize this relationship in Equation (10):(10)$$C\approx \Delta \left(SEP\right)\cdot \Delta \left(MEP\right)$$

Equation (10) represents a tradeoff between SEP and MEP. It indicates that a team cannot fully know both SEP and MEP simultaneously. However, Equation (10) also indicates that as uncertainty in the structural entropy production (SEP) decreases to a minimum, the team is able to devote more and more of its available free energy to maximizing its performance (MEP).

In the literature about the inevitability of the need to resolve conflict for an organization to be coherent [75], coupled with the example from the field about J&J’s replacement of its Emergent contracted manufacturer (i.e., [76]), Equation (10) also indicates that achieving structural fitness is key if a team or organization is freely able to choose the best available candidate to be a member of the best performing team, reducing entropy in a way that is similar to adding the last element to complete the structure of a molecule (e.g., when ions of sodium and chlorine combine to form a crystal of salt, their combined entropy drops). What happens when a well-trained team member is replaced by a highly capable but new teammate? Initially, structural entropy increases dramatically while the newcomer builds relationships with existing teammates, but as the new teammate settles in and masters the role of the individual replaced, the team’s structural entropy production drops, allowing the team’s performance to recover [50].

On the other hand, by inverting this concept, according to Equation (10), when a team increases its structural entropy production for whatever reason, all things being equal, the team must correspondingly reduce its maximum entropy production. What could lead to such a reduction? Of all of the possible causes, we focus on a vulnerability discovered in one’s own team structure, leading to a search for new candidates among the best available for a merger; a vulnerability created in a team by an old merger; a vulnerability created purposefully during an attack; or a vulnerability discovered in an opponent, leading to the exploitation of the opponent.

## 7. Research Design

First, we will review mergers and how vulnerability affects firms. Second, we will revisit the most recent MENA data with a correlation matrix to provide insight into vulnerabilities in nations in the MENA countries (the UN designation for Middle East North African countries).

## 8. Results

### 8.1. A Hypothesis from Case Studies of Vulnerability in the Field

To illustrate the vulnerabilities posed by Equation (10), we first look to the literature, and then, the field to find examples of teams (firms, organizations, alliances and systems) that fit into one of three categories of mergers to reduce a vulnerability in a business, to spin-off a previous merger that has failed, or to exploit a vulnerability created in another business or organization.

#### 8.1.1. Mergers Designed to Prevent a Vulnerability in Being Able to Compete

According to the literature [79], mergers inform us about teamwork, cooperation and competition; about how systems and alliances may function; and about how well markets are performing. Alliances can form among rivals when facing emergencies, such as national defense or the COVID pandemic in 2020–2021 (for alliances promoting national defense, see [80]). In the field, mergers may reflect a vulnerability from a lack of the technology to compete; of the assets and infrastructure needed to address an overwhelming problem; or of the need for scale to cope with an international emergency, such as the development of new vaccines (e.g., [81]). Mergers can be used to enter new markets. For example, Allied Universal’s bid for G4S PLC was designed to create a private security-services giant that would strengthen Allied’s operations in the U.S. and the U.K. while extending its operations in Europe, the Middle East and Africa [82]. Mergers may be used by a firm to strengthen its competitiveness after gaining access to new technology, such as Northvolt’s acquisition of Cueberg to increase the range of its electric-vehicle batteries [83]. A merger can also be designed to save on expenses while preparing to invest in the adoption of a new technology, such as the 5G wireless service business in Canada [84], or can be pursued to expand services, such as DoorDash’s purchase of Chowbotics, a robotic fresh-salad maker [85].

In the literature, once a vulnerability is discovered by a firm in itself, action is taken through the mechanism of a merger to offset its perceived weakness [79]. In the field, the same step is taken whenever leaders are thought to be the problem, e.g., the change in managers at Credit Suisse after their involvement in the collapse of the Greensill Capital supply-chain finance funds [86].

#### 8.1.2. Spin-Offs to Reduce Vulnerability

In the literature [87], not all mergers or acquisitions work. In the field, AT&T purchased DirecTV near the peak of the market for payTV, which ended shortly thereafter due to the cord-cutting movement; today, AT&T split off this same unit, which has struggled and lost significant value since it was purchased [88]. After International Business Machine’s (IBM) artificial intelligence system “Watson” bested humans on a quiz show, it indicated a possible new business in health care that has not lived up to its hype [89]: “Watson Health has struggled for market share in the U.S. and abroad and currently is not profitable.” Since competition has tightened returns on solar plants for what was “once the world’s most ambitious solar investor”, SoftBank Corporation is now selling its renewable energy firm to a rival, Adani Green Energy Ltd. [90]. With its investors frustrated with the company’s performance [91], Dell plans to “spin off its stake … in cloud-software company VMware … to strengthen its financial position as it looks to enter new markets and works to reach clients revamping their digital strategies.” New management at Citigroup has motivated the bank to reduce its competitive vulnerabilities by “moving to exit consumer banking franchises in 13 countries in Asia, Europe and the Middle East” [92]). Reflecting the poor situation currently existing for European banks, HSBC Holdings PLC in the UK is planning to sell its French retail bank for EUR 1, a significant loss for a bank it had bought for over USD 10 billion just two decades ago [93].

The DuPont Company was once the largest employer and philanthropist in the state of Delaware, funding schools, libraries and theaters [81]. Joe Biden, now President, celebrated his victory for political office in 1972 in the Hotel DuPont. “More than four decades later … DuPont, struggling to boost profits, was targeted by an activist shareholder, sold the hotel, eased out its chief executive, merged with another company, split into three pieces and cut its Delaware workforce by one-fourth [94].”

#### 8.1.3. Exploitation of Vulnerability: Deception

Equation (10) gives guidance on how a machine, as part of a team, can blend in with its teammates should its function be to deceive its teammates, as the deceiver seeks to create a vulnerability in a team, firm or system. Using deception, an individual playing an instrumental role in a team could double as a spy against the team. To keep from being perceived as a “bad” actor within the team by fellow teammates, based on Equation (10), a deceiver must reduce the entropy it produces structurally, making it particularly difficult in cyber-warfare to identify an unwelcome intrusion [95] when an interloper does not stand out (e.g., in the literature, [82]). An example from the field is what happened with Volkswagen [96], an organization selling cars by deceiving the public and the public’s regulators regarding its faithful application of the rules, a deception that succeeded until it was uncovered years later; namely, six years ago, Volkswagon admitted to the deceitful gambit that its emissions failed to meet regulatory standards [83]: “U.S. authorities charged Volkswagen with conspiracy to commit fraud, making false statements on goods brought for sale in the U.S. and obstruction of justice. The U.S. probe uncovered a decade long ploy by Volkswagen to rig millions of diesel-powered vehicles to cheat emissions tests and later attempt to cover up the cheat.”

In the literature, dependencies can become vulnerabilities (e.g., in cyber, see [97]). In the field, the fight between Apple and Facebook [98] directly addresses the exploitation of a created vulnerability [84]: “For the past few weeks, Facebook has been running an ad campaign in defense of personalized advertisements, arguing that targeted ads are key to the success of small businesses. The catalyst for the campaign has been an ongoing battle between the social media company and Apple. The battle focuses on a unique device identifier on every iPhone and iPad called the IDFA. Facebook and others that sell mobile advertisements rely on this ID to help target ads to users and estimate how effective they are. … apps that want to use IDFA will have to ask users to opt in to tracking when the app is first launched. If users opt out, it will make these ads a lot less effective. Facebook has warned investors that these looming changes could hurt its advertising business …” In the Fall of 2020 during the height of the pandemic in the United States and the election season, Southwest Airlines seized the opportunity to exploit the struggles of its competitors [85]. “The pandemic is forcing many airlines to defend their turf. Southwest is using it to invade. Even as air travel languished in this fall, Southwest Airlines Co. executives fanned out to cities from Palm Springs, Calif., to Sarasota, Fla., to scope out potential new markets. The airline is adding four more cities to its network this year and announced plans for another six in 2021.”

### 8.2. Results of Correlational Tests for Theses of Vulnerability in MENA Countries

Theses: The following theses focus on vulnerability. Based on Equation (10), should a team become vulnerable (characterized by a higher SEP), it is likely to struggle with defending itself and with being more productive.

Thesis 1: As education (HDI) improves across a population, a nation becomes more intelligent and less vulnerable, reflected by its patent productivity increasing. The results in the CM in Table 1 below support the hypothesis; as HDI increased, patent productivity rankings in the MENA countries increased significantly (r = −0.81, *p* < 0.001).

Thesis 2: As freedom scores increase and teams are freer to replace underperforming teammates (Equation (10)), a team’s improved fitness leads to less vulnerability and an increase in patent productivity rankings (PPR). The results in the CM below support the hypothesis (Table 1); as the freedom ranking of a MENA nation increased, its patent productivity rankings increased significantly (r = −0.60, *p* < 0.02).

Thesis 3: As a population increases, an increase in redundancy (the number of free riders) reduces its ability to evolve, reflected by its lower patent productivity rankings (PPR); the results (Table 1), while not significant, agree with the hypothesis in that as the population increased in a MENA country, its patent productivity rankings decreased (r = 0.28, *p* n.s.).

Thesis 4: As corruption perceptions (CPI) increase across a society, if they are not sufficiently free to root out the corruption or deception, its ability to evolve decreases, reflected by a decrease in its patent productivity rankings (PPR). The results of the association in the CM support this hypothesis (Table 1); as CPI scores decreased in the MENA countries, their patent productivity rankings decreased (r = −0.76, *p* < 0.001), freedom decreased (r = 0.55, *p* < 0.05) and population levels increased (r = −0.48, *p* n.s.).

Other findings: In the MENA populations surveyed in Table 1, as their populations increased, freedom scores decreased (r = −0.70, *p* < 0.01) and HDI scores also decreased, but not significantly (r = −31, *p* n.s.). For these populations sampled in the MENA countries, we also found that Gross Domestic Product (GDP) was not significantly related to HDI scores (r = 0.24, *p* n.s.), freedom scores (r = −0.47, *p* n.s.), patent productivity (r = −0.17, *p* n.s.) or CPI scores (r = 0.06, *p* n.s.).

## 9. Discussion

While we recognize its importance in decisions, we discuss time and its effects on the interaction from a theoretical perspective briefly and later (e.g., [99,100,101,102,103,104]).

### 9.1. Mergers

We have made several points. First, in support of causality, Pearl [42,43] has been forceful in communicating the need for human and machines in a team to be able to communicate clearly in a language that both understand, which was certainly violated in the Uber self-driving car fatality of a pedestrian [24,44]. The Uber car’s failure to communicate with its human operator strengthens the value of interdependent communications among human and machine teammates to let each other know when either perceives a threat.

Second, and in support of Bohr’s [15,17] theory of complementarity, we adduced equations as a way to organize the plentiful evidence from the literature and in the field, in support of Equation (10), which states that tradeoffs exist in the struggles of humans to survive when confronted with uncertainty, conflict or both.

Third, during a competition between two evenly matched teams, if an attack finds a vulnerability that triggers a tradeoff between an opposing team’s structure and its performance (i.e., in the literature, dependencies create vulnerabilities [97], e.g., the vulnerability of Facebook exposed by Apple’s change in its new app-store rules; in [105]), excess structural entropy characterizes the vulnerability [46]; this is a discovery that both machines and humans can mathematically use to determine a weakness in an opponent, or in one’s own team. Additionally, as an example of coherence, protecting a team’s vulnerability occurs by trusting its agents [106], but if they are in orthogonal roles, orthogonality reduces the confirmatory evidence when those roles work. Should these two results be validated in the field, this suggests how the defeat of an opponent [107], or the remediation of the weaknesses in one’s own team, is enacted, but indirectly, by reducing the intrinsic value of trust.

Fourth, as interactions become less restricted (e.g., a free market [93]), Equation (10) represents the tradeoffs that occur between structure and performance. To exploit a people by making their survival contingent on vulnerability, the blocking of tradeoffs under oppressive leaders accounts for how gangs, authoritarians and monopolies construct barriers to growth, wittingly or not; however, by suppressing alternative viewpoints, the very key to innovation [52], these regimes become hampered by the suppression they inflict on others when they need to innovate, especially as happens when facing uncertainty or conflict.

Regimes of oppression can innovate by using deception to steal ideas [108]. For example, although praised by Charlie Munger, who stated that “China has managed to lift about 100 million impoverished people out of extreme poverty since 2012,” (https://news.cgtn.com/news/2021-02-26/Charlie-Munger-staggered-by-China-s-poverty-elimination-success-YbFCa6WjyE/index.html (accessed on 15 August 2022)), enforced cooperation in China has led to its vulnerability and its need to steal technology (e.g., [109,110]). China’s widespread political suppression has impaired its ability to innovate, leading it to hide new developments, e.g., according to the *New York Times* [111], “Sinovac, which is produced by a Chinese pharmaceutical company and promoted by the Chinese government, has been criticized for a lack of transparency about its clinical trials.” Similarly, by rejecting the latest offer for the “establishment of permanent peace on the Korean Peninsula” [112], North Korea continues to trap its own people; unable to feed or to help themselves, many of them are tricked into slavery by promises of a “Socialist Paradise” [113].

Not surprisingly, the inability of China to innovate with a new vaccine also occurs with business monopolies. In the U.S., the company Amazon, being a monopoly, has created its own vulnerability from its lack of innovation, motivating its need to steal ideas from some of its best clients [114].

### 9.2. Discussion: Correlation Matrix for the MENA Countries

As a team’s fitness improves, as in Equation (10), the improvement is characterized by a reduction in its structural entropy production (SEP) along with a complementary increase in the team’s productivity (MEP); this is best illustrated in the MENA region by the technological leadership of Israel. Team fitness, however, depends on a team’s ability to freely choose its members in order to reduce its vulnerabilities, to increase its strengths and to learn from its mistakes, consequently advancing its social evolution. The primary reduction in the vulnerabilities of a team is the team’s reduction in redundancy among its team members (free riders). The key to understanding this aspect of team size is that teams work best when interdependence is allowed to reach a maximum [67].

Freedom allows a team to self-organize to minimize its SEP (a reduction in free riders and the ability to freely choose a member’s replacement over a forced choice), leading to an increase in its evolutionary competitiveness, reflected by a complementary increase in MEP.

In contrast, in agreement with Equation (10), authoritarian regimes depend on social compliance; they do not fear corruption getting in their way, but instead, exploit it to assist with social compliance by forcing the family members or associates of key allies to be hired by productive teams, increasing a team’s level of redundancy, but thereby reducing its productivity. The key to understanding corruption administered in the lower levels of a corrupt hierarchy, business or government agency (e.g., the military; [51]) is knowing that redundancy helps to increase social compliance by reducing social turmoil, causing a reduction in evolutionary pressure.

Complementary tradeoffs by authoritarian leaders are evident in the results. In agreement with Equation (10), the citizenry in an authoritarian regime must be controlled, even if that means impeding the primary path to innovation available through free markets and free speech (Lawless, 2019); the tradeoff is that increased SEP costs reduce a team’s performance outcomes by reducing its MEP, but, across an authoritarian society, the ability of a country to evolve becomes compromised.

## 10. Future Research

First, many years after Bohr, but clearly capturing Bohr’s concept of the tradeoffs captured by complementarity, Cohen [100] described signal theory as transformations between Fourier pairs and that a “narrow waveform yields a wide spectrum, and a wide waveform yields a narrow spectrum and that both the time waveform and frequency spectrum cannot be made arbitrarily small simultaneously” (p. 45).

In this study, we have focused on complementarity and causality, open science and interdependence and the effects that machines can discern. Equation (10) tells us, however, that the arrow of time [101] for the perfect team in action should be harder to link causally to the actions performed by the team. Supported by the literature, this appears to be the case for the perception of time: “relatively passive activities appear longer than do those requiring active participation” [102]. Moreover, “if causal asymmetry is mind-dependent … then we cannot appeal to it in accounting for our experience of temporal asymmetry—the difference between past and future” [103]. To the point and for further investigation, in Rovelli’s *The Order of Time* [104], time is the direction in which entropy increases for a closed system from which we gain information [101]. Regardless, if the perception of time is obscured during a performance by a perfect team, it suggests that the structures of perfect, the best or even good teams (marriages, mergers, alliances and human–machine teams) depend on trial and error, and a successful match, identified using Equation (10), is characterized by a reduction in structural entropy as a team is assembled. As we previously mentioned, this speculation is supported by the National Academy of Sciences, who claim that the “performance of a team is not decomposable to, or an aggregation of, individual performances” (p. 11 [14]).

Second, we are pursuing metrics for the physics of the interdependence between the structure and performance of teams (Equation (10)). How much is a team’s productivity amplified versus the individuals who compose it? What happens to an autonomous team or system when issues arise unexpectedly such as confronting contexts of uncertainty, conflict or competition? Additionally, how might the governance of an A-HMT-S be conducted? Namely, how much authority should be given to human or machine members in an interdependent team to override human or AI counterparts in the event that an operator or leader becomes dysfunctional?

Third, of its recommendations that we plan to pursue, the NSC—Artificial Intelligence (AI) *Final Report* (p. 12, https://reports.nscai.gov/final-report/table-of-contents/ (accessed on 15 August 2022)) not only motivates human–AI teaming for national security, but is also a model to increase user and public faith in AI. We believe that our model of the adversarial pursuit of context agrees with Ginsburg [61], in that it could increase transparency by testing vulnerability in opposing teams or in one’s own team, and it could promote human–machine governance [20,46].

Fourth, one of the problems with machine learning that we also plan to study is its lack of generalizability [9] from one context to another [11]. However, with the ability in the future of machines to learn using one-shot learning (reviewed in [10]) combined with prior learning, and the more generalizable AI combined with complementarity, we believe that machines, with human help, may be able to reprogram themselves on the spot, overcoming present design limitations regarding the determination of context. This concept addresses what we see as von Neumann’s [115] problem with the aggregation of self-reproducing automata.

Fifth, in the future, we hope to address a limitation of Shannon’s information theory with its inability to serve as a theory of information value. Suppression in China and North Korea to achieve a stable society (simulated using NetLogo’s Ising model initiated with spin-ups at 100% and low social temperature, evolving over time to a mix), contrasts starkly with the approach taken by freer societies where humans struggle adversarially to determine uncertain contexts. The need for an “informed assessment of competing interests” that was recognized by Justice Ginsburg ([61], p. 3) exemplifies the processing of interdependent (bistable) information. Ginsburg’s decision implies that a context facing uncertainty can only be resolved interdependently by teams in competition or conflict [41]. We speculate that Equation (10) points to a theory of value information processing.

Sixth, and for a future project, we plan to apply Equation (10) to address minority housing. We know from Equation (10) that structure is important, but it also tells us that to “fit” in means to not emit excess structural entropy, suggesting that deception, while sometimes a curse, can be a cure; for example, the Electronic Benefit Transfer (EBT) card, which looks like a credit card, allows minorities on welfare to avoid stigma, when buying groceries, caused by the embarrassment of using Food Stamps in a public transaction (https://www.fns.usda.gov/snap/short-history-snap (accessed on 15 August 2022)).

## 11. Conclusions

First, from the Correlation Matrix, in Equation (10), a complementarity exists between a team’s structural entropy production (SEP) and its performance, characterized by its maximum entropy production (MEP). If a team has internal conflict (e.g., a prior merger’s failure; divorce; infighting; or power struggles), a lack of motivation (e.g., social suppression by an authoritarian regime or a command economy) or even a lack of ability to make a claim for the ownership of a patent, its productivity will suffer.

Second, if it can be shown that the mathematical physics of autonomous human–machine teams is not only possible in theory, but also applicable in the field, the potential impact for our work-in-progress on the development of a new direction for the science of interaction would be profound. Directly applicable to an A-HMT-S, it would allow human–machine teams to determine, by observation, whether an opponent’s team was performing at a maximum level, was malfunctioning or was vulnerable and how to exploit it; alternatively, it would enable the determination of whether its own team was vulnerable and in need of a trial-and-error process to reduce its vulnerability by retraining or replacing an existing member.

The human brain versus the mind: Gazzaniga’s [56] study of split-brain patients led him to discover that “the left half did not know what the right half was processing” (p. 57). Brincat et al. [57] concluded the same in research with visual working memory as part of a seamless whole. This seamless combination of information from the left and right brain may operate to determine reality similarly in the social dimension, where information passes between one debating team to another, or from the prosecuting attorney to the defense attorney to the jury and the judge, with ideas flowing from one side of a debate and triggering a response from the other; each side is able to identify the key idea summoned by the winning side, but is sometimes unsure of how it may have arisen or was prompted; they stitch reality together, underscoring the value of the bicameral U.S. Congress and the checks and balances afforded by the U.S. Constitution, but also highlighting how an audience makes causal sense while immersed inside of a fast-moving narrative as the key idea in it takes shape and unifies it into a coherent whole [101].

Third, causal sense is important in building a narrative over time for a story, a movie, or a communication between a machine and its operator. However, what happens when uncertainty or conflict beclouds an ongoing narrative? Then, among free humans, a unified narrative is spontaneously displaced by debate between bistable, opposing viewpoints in a search for the least vulnerable path going forward [46]. Bistability has multiple interpretations [55]. However, the brain only “sees” one interpretation of bistable reality at a single moment in time [54]. Based on the model by Gazzaniga [56] stating that the two halves of the brain are independent, and by Brincat et al. [57] stating that the independent halves of the brain stitch together sensory information into a coherent view of reality, it is important for AI to address this process further, and it has. AI has recently become engaged with champion debaters, in “which humans still prevail,” but the AI is improving over time [102].

Lastly, in conclusion, the science of autonomous human–machine systems has to be open, applicable in the field and practical; however, once a discovery has been made in the field, it should be replicated in the laboratory and used to build new theory [23]. While traditional causal interpretations of reality may suffice in known contexts (roadways; airports; mining operations, etc.), we have challenged the traditional model of interaction in situations where autonomous teams are facing uncertainty or conflict with debate narratives (e.g., see [116,117]), and in the rational design of the best human–machine teams. By separating team structure and performance, we extended our theory, borrowed from Bohr, of interdependent complementarity to propose a new direction of research. With this, we discovered that competition between evenly matched teams hinges on vulnerability, characterized by excess SEP and reduced MEP, which we believe generalizes to autonomous human–machine systems. We conclude that Shannon entropy generated by independent and identically distributed (i.i.d.) data does not generalize; however, interdependent complementarity does. This supports our call for the new science of autonomous human–machine teams and systems [118].

To generalize, autonomous human–machine teams operating in a less free system will be vulnerable to command decisions, less fit to make decisions in the interests of their society, at an evolutionary disadvantage and less able to step in to take over when a human leader becomes dysfunctional (e.g., when a copilot is attempting to commit suicide).

## Figures and Tables

**Figure 1 entropy-24-01308-f001:**
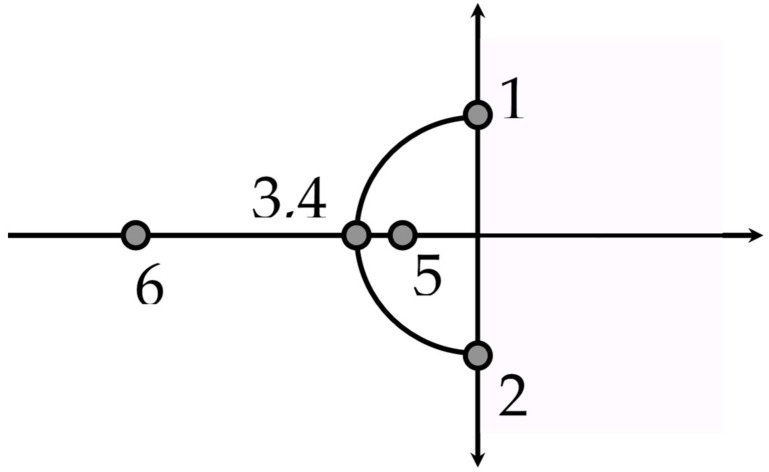
The *y* axis or imaginary axis represents the beliefs held not in physical reality, but instead, in the minds of debaters, and re-constructed in the collective minds of an audience. When no decision for action is forthcoming, especially when there is no audience and the debate is endless, the debate is located at points 1 and 2; in the extreme, it is observed as a “war of words” [68]. The *x* axis represents physical reality (e.g., action). Often encouraged by an audience (e.g., a jury), when debaters reach a compromise for action, that decision is represented by points 3 and 4. When debate is stopped by resistance from the audience with a decision acceptable to the majority in an audience, that decision is reflected by points 5 and 6 ([20,46]).

**Figure 2 entropy-24-01308-f002:**
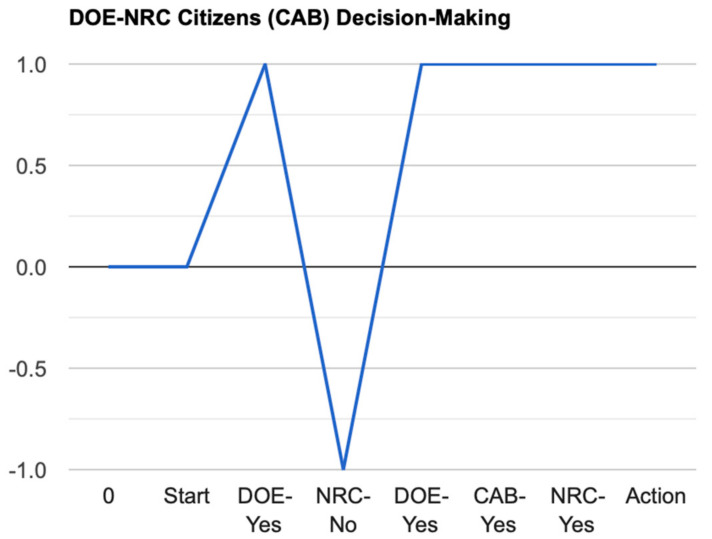
Rotations over time (compressed) to reflect the NRC’s decision to allow the DOE to proceed based on an orthogonal rotation of positive feedback, then negative, then positive, etc., which continued for almost 7 years until the NRC conceded to a “yes” decision. Since then, six more tanks have been closed safely under regulatory oversight (https://www.srs.gov/general/news/factsheets/srr_radioactive_tank_farms.pdf (accessed on 26 June 2022)).

**Figure 3 entropy-24-01308-f003:**
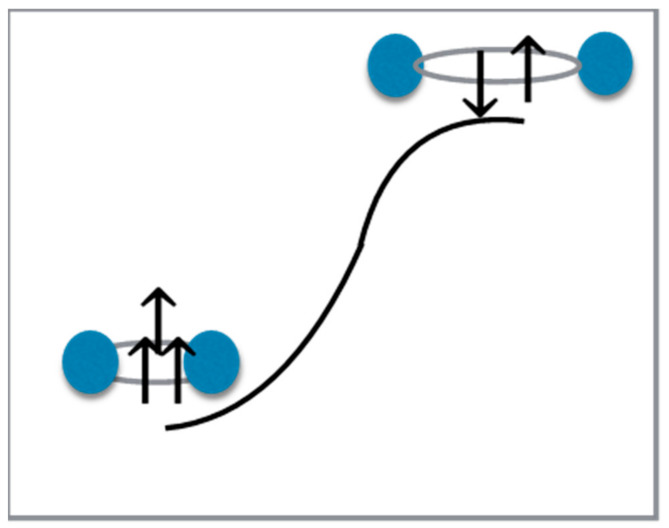
Shown in the lower left in this figure is the ground state for SEP (Equation (5)). Shown in the upper right is the excited state for SEP (Equation (9)). In the ground state for SEP, an extra arrow denotes that the interference flowing through the team produces the team’s more productive state. For the excited state, however, all of a team’s free energy is being directed at tearing apart the team’s structure, maintaining its interference in a destructive state.

**Table 1 entropy-24-01308-t001:** The correlation matrix (CM) calculated from the data for the Middle Eastern North African (MENA) ^a–d^ countries (the levels of significance for all two-way associations were determined using two-way Pearson correlations, resulting in a probability, *p*, of chance set of less than 0.05 for correlations of 0.51 or above; less than 0.02 for associations of or above 0.59; 0.01 for correlations of or stronger than 0.64; and 0.001 for those of 0.76 or greater; for this data set, N = 15, giving degrees of freedom of 13).

Valences of Factors (in Parentheses)	HDI	Freedom	Population	PPR	GDP	CPI
Human Development Index (HDI; higher) ^a^	1					
Freedom (more) ^c^	0.53	1				
Population (greater) ^b^	−0.31	−70	1			
Patent productivity ranking (PPR), (inverted) ^b^	−0.81	−0.60	0.28	1		
GDP (greater) ^b^	0.24	−0.47	0.70	−0.17	1	
Corruption Perceptions Index (CPI; inverted) ^d^	0.76	0.55	−0.48	−0.76	0.06	1

Notes on Sources: ^a^. Human development insights; education: https://hdr.undp.org/data-center/country-insights#/ranks (accessed on 15 August 2022). ^b^. Patent productivity rankings (PPR); GDP; population (five MENA countries were omitted due to their lack of a patent productivity ranking: Iraq; Libya; Palestine; Sudan; Syria): https://www.globalinnovationindex.org/Home (accessed on 15 August 2022). ^c^. Freedom Index: https://worldpopulationreview.com/country-rankings/freedom-index-by-country (accessed on 15 August 2022). ^d^. Corruption Perceptions Index: https://www.transparency.org/en/cpi/2021 (accessed on 15 August 2022).

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
