# Peer review of "Interdependent Autonomous Human–Machine Systems: The Complementarity of Fitness, Vulnerability and Evolution"

_entropy, 2022, doi:10.3390/e24091308_

Round 1

Reviewer 1 Report

(1)  The author proposed the Structural entropy production (SEP) and maximum entropy production (MEP). However, there is no real experimental data to verify the feasibility.

(2) The paper lacks necessary theoretical model diagrams, tables and other descriptions.

(3) The author cites a large number of references in the paper, but it is only a simple expression, and there is no detailed analysis and description of it.

Author Response

Dear reviewer:

I hope that my responses to you have adequately addressed your concerns. Should you need additional information or changes, please let me know.

Reviewer 2 Report

It is an interesting paper. In reviewer’s opinion, the author presents an interesting work and the readers will find it useful. The subject of use of entropy for human-machine systems is relevant. The topic is fascinating and relatively new, but the paper can be improved on different aspects.

The core issue reviewer finds in this work is that the validation the proposed framework is relatively poor. Reviewer thinks that more results have to be presented to demonstrate the potential of the proposed framework. Reviewer also felt that higher level of validation for a “Article” type paper is needed. Indeed, reviewer thinks that this paper suits better for a “Communications” type submission.

Some other general notes:

The actual contribution of the authors to the proposed methodology should be explained more clearly.

Abstract too long, it can be shortened.

Figure captions are too long, text in figure captions can be moved into main text.

Too many footnotes, try to avoid footnotes, put them in references instead.

Author Response

(The authors gave the same response as above.)

Round 2

Reviewer 2 Report

Reviewer appreciate the author’s responses. Many points have been clarified. Still, it’s not clear, even with additional correlation matrix, validation of the proposed system. It should be more clearly described and discussed.

Author stated that figure captions, abstract are shortened, which are not shortened, and author stated number of footnotes have been reduced, where there are still 9 footnotes in total.